# Tablet-Based Praxis Developed for Children in Primary Education Studying Natural Sciences and Mathematics

**DOI:** 10.3390/children10020250

**Published:** 2023-01-30

**Authors:** María-Carmen Ricoy, Cristina Sánchez-Martínez

**Affiliations:** Department of Didactics, School Organization and Research Method, Faculty of Education and Social Work, Universidade de Vigo, 32004 Ourense, Spain

**Keywords:** tablet, blog, primary education, natural sciences, mathematics education

## Abstract

Information and Communication Technologies are now a common feature in classroom activities. The aim of this study was to present praxis developed for the tablet for use by primary education students (aged 6–12) studying the natural sciences and mathematics. This research is qualitative and follows the narrative-ethnographic approach. The study sample consisted of 120 primary education students and 52 educational blogs. The results and conclusions reveal praxis that is rarely innovative or ludic. The bulk of tablet-based activities were for natural sciences classes rather than mathematics, and the most common practice with the tablet in the natural sciences was information searching and content exploration. The most widely used apps were the Google search engine, YouTube and the tablet’s default apps (camera, image and video editor). Course content in the natural sciences focused on living beings and states of matter, and the activities developed for children to do on the tablet aimed to foster learning through discovery, exploration and enquiry. In mathematics, a traditional methodological approach was apparent in children’s use of the tablet for typical activities related to units of measurement.

## 1. Introduction

Information and Communication Technologies (ICT) are everywhere in our daily lives, and are increasingly used in the classroom. Tablets have become more visible and more relevant to school activities, in part due to their extensive use by people everywhere. School centers have adopted them as an extension of their widespread use at home [1]. The main advantages of the tablet are its mobility, small size and lightness, which give it an advantage over other mobile devices for classroom use [2,3].

ICT, and the tablet in particular, have considerable potential and are very popular as resources that are attractive to all, especially the young. Legislation on education in countries around the world calls for students to acquire digital competences, and Spain is no exception [4], stressing the importance of developing competences in mathematics, the Sciences and Technology; for this reason, mathematics and the natural sciences form part of the primary education curriculum. Similarly, governments worldwide are keen to foment digital literacy and the acquisition of high-quality skills in the Sciences among students in order to boost economic, social and environmental development [5]. The Organization for Co-operation and Development (OECD) defines literacy in science as the ability to engage with the Sciences through related activities in a critical and reflective way [6].

The new technological resources can help children to overcome the difficulties they may encounter when learning the natural sciences and mathematics. Research has shown that the use of digital tools can improve academic performance by involving students more in their learning, in particular of mathematics and the sciences [7]. For this reason, children’s teaching-learning processes in the natural sciences should be supported through practices that develop problem solving, the creation simple experiments, skills acquisition and the management of critical thinking [8,9].

In mathematics, the scientific literature proposes that classroom activities include a range of tasks that stimulate students to ask questions, solve problems, explore, analyze data and documents, experiment, formulate conjectures, create representations, etc. [10]. Education researchers highlight the importance of practices that develop problem solving in the teaching and learning of mathematics as essential components of the school curriculum [11].

In order to ensure better science literacy among children in the future, starting with their learning in the natural sciences and mathematics, it is vital to develop didactic practices that are supported by digital technology [12,13], and that can be developed and driven by mobile devices such as tablets. Science education contributes to the improvement of the understanding of science and the levels of scientific literacy in society [14]. The ultimate aim is to underline the importance of the Sciences and educate children as they develop in the digital era through the use of the latest technological resources.

The adoption of technological resources such as the tablet in the classroom enhances students’ commitment, motivation and attention, even among children who find learning mathematics a challenge [15,16]. Hence, mobile devices in schools have now become an essential component of student learning [17,18]. Students using the tablet can rapidly access course content, and can alter and edit it easily [19,20]. Tablet usage encourages ubiquitous learning inside and outside the classroom. Children generally need to be presented with learning experiences to improve their efficacy and stimulate their creativity when doing classroom tasks [21,22], so digital mediation can improve the learning process by clarifying the definition and perception of course content.

The potential for doing schoolwork on the tablet is vast, especially with the wide range of digital apps available [23,24] that can provide multimedia and ludic resources that enable students to carry out different tasks [25,26]. Apps are tools that offer a variety of possibilities that are very attractive for students to work with, both in the classroom and at home. Such resources provide lots of alternatives for use in the course work for the natural sciences and mathematics.

Studies have examined the use of tablets in these two subjects and found that, for example, students using the tablet acquired a solid procedural and declarative knowledge of plants [27]. Likewise, a positive impact was observed in students’ motivation to learn in young children. The use of augmented reality with tablets enabled students working on course content related to endangered animals to increase their levels of concentration on the subject [28].

When the tablet is used in mathematics classes, in combination with the appropriate software, it can greatly increase students’ understanding of numbers [29]. For example, children used a digital map on the tablet to track a set of pre-selected places in a park in order to resolve Mathematical problems using data from the setting [30]. This study found that the students solved puzzles by applying mathematical knowledge and through discussion and collaboration with classmates when gamified activities were used to reinforce their learning.

In general, children who use mobile devices such as the tablet for learning achieve significantly higher levels of literacy in science compared to those who stick to traditional resources [31,32]. However, the impact of the use of mobile devices on learning remains unclear, as the results of studies tend to vary. Research on tablet use in the classroom continues to be limited and fragmented [33], even more so in the Sciences and mathematics [34,35]. This makes it essential to analyze cases in context to broaden our knowledge of the effects of tablet usage in these two academic areas, and to understand how to develop innovative, didactic, or traditional activities that can be tablet-supported. It is important to analyze the experiences developed by using technology to determine whether they can be transferred to other educational contexts.

The main aim of this study was to discover the praxis developed for the tablet in the subjects of the natural sciences and mathematics for primary education children aged 6–12. The research questions (RQ) were:− RQ1. What typology of activities was used with the tablet in the natural sciences and mathematics?− RQ2. What apps were used with the tablet, and how, to support activities in the natural sciences and mathematics?− RQ3. What was the curricular content that underpinned the activities carried out on the tablet in the natural sciences and mathematics?

## 2. Methodology

This research was qualitative, with the potential for knowledge construction based on the study of multiple cases. Identification, description, analysis, interpretation and explanation in this study all helped to enhance our understanding of the reality of the situation [36]. The narrative-ethnographic approach adopted in this research enabled an extensive examination of the data, delivering knowledge of teaching practices based on new ways of gathering information [37,38]. Technology was also used to share, disseminate and promote a range of experiences among citizens, professionals and students.

### 2.1. Procedure, Data-Gathering Techniques and Samples

This study was developed using two data-gathering techniques: group discussion and the analysis of digital documents (blogs). The group discussions were set up to study multiple cases that included students in different classes in primary education centers. This enabled the real protagonists (the children) to voice their opinions on the use of the tablet. Three primary education centers in northwestern Spain were used. The sample selection was deliberate, in that those selected had adopted the tablet as a support tool in the learning process, which was the focus of this research.

The qualitative nature of this investigation meant that it centered on students across the spectrum of the six grades of the primary education stage, studying multiple cases in four grades in particular (1st, 3rd, 5th and 6th). A total of 120 students were selected (56 boys, 64 girls), with an age range of 6–12. The questions posed to the children in the group discussions were aimed at discovering:− The activities they did on the tablet;− The apps used on the tablet, and how they used them;− The typology of the course content developed by the activities done on the tablet.

The data were gathered from 24 group discussions at the school centers. Each of the four primary education grades were organized in 3 groups of 5 children, with each grade’s groups being called to meet on successive days. The dialogue with and among the children was led by a researcher with experience in this technique and subject area (the researchers in this study). The conversations in each group lasted between 20 and 30 min.

At the same time, a sample was assembled of blogs that discussed experiences of working with a tablet in primary education. This was accomplished by searching Google to locate the main online education blogs. An initial data approximation involved tracking key words such as “blog”, “educational” and “ICT” in the online search and discovering websites that aggregate lists of blogs.

A total of 52 blogs were selected as the definitive sample for the analysis of the digital documents; the inclusion criteria were:− Education blogs centered on primary education that discussed experiences developed in Spain.− Blogs that disseminated experiences that used the tablet in the natural sciences and/or mathematics classes in primary education.− Blogs written strictly by primary education teachers in Spain.

### 2.2. Data Analysis

All of the data collected were organized prior to the analysis of the information from the blogs and group discussions. The analysis of the content required these sources of information to be treated individually. The definition of the main units of analysis was established by five experts from three Spanish universities, each a specialist in the study object and in content analysis. Based on their deliberations, the first-level categories were grouped ad hoc according to the raw data, considering their link to the general objective of the investigation and the three RQs [39] (Table 1).

Content analysis was performed using the Analysis of Qualitative Data (AQUAD) software, which coded the information in the texts from the group discussions and digital documents (blogs). This procedure ensures that the coding of the data is based on a solid interpretation [40]. The content analysis was carried out by pairs of researchers with a strong background in this type of analysis, and whose coding complied with the initial premises indicated by the experts. They first agreed on the type of coding to be applied and then resolved any minor discrepancies that arose in the final phase.

The transfer of the results from AQUAD to Excel helped to structure the results, and a process was executed to relate the categories and subcategories (codes) resulting from both data-gathering techniques (group discussions and digital documents). The strategies applied in the content analysis enabled the presentation in the Results section of the (absolute and relative) frequency of the respective codes obtained, which is an interesting contribution for discovering potential preponderances or trends.

## 3. Results

The results of this study are presented in graph form and text, and, to optimize the organization of the results, the main categories are aligned with the research questions, taking into account the relation between the nucleus of the analysis and the subjects of the natural sciences and mathematics taught in primary education (Figure 1).

Prior research on group discussions showed that the natural sciences was the subject for which the greatest number of activities had been developed for tablet use, as opposed to mathematics, where there were far fewer initiatives involving tablet use. The present results also reflect the analysis of the digital documents in the form of blogs, which are presented in depth across a range of sub-sections. In this way, the results for the praxis developed with the tablet reveal sharp contrasts between these two subjects in the school curriculum.

### 3.1. Activities with the Tablet Supported by Apps (RQ1 and RQ2)

The practices developed by children using the tablet in the natural sciences and mathematics classes in primary education relate to a range of typologies. There was a difference between the activities generated in the school centers, as analyzed in group discussions with the children, and those described in the blogs. Nevertheless, patterns emerge among activities designed for tablets for both subjects, as shown in Table 2.


*Students in the 5th and 6th grade at school center XXX have worked on “R for Reduction”, gathering information and creating slogans to raise awareness on reduction, as well as producing drawings to accompany the slogans, using Snote on their tablets (Blog nº 7).*


Natural sciences classes saw other patterns of activities supported by the tablet that fomented children’s learning through discovery, exploration and enquiry (GD = 40/120, 33.33%; Blog = 30/52, 57.69%). This typology of activities emerged from the development of short educational projects in which the children located information then analyzed and interpreted it, encouraging them to ask questions and make predictions and proposing responses and explanations. This praxis, though not common, predominates in the experiences described in the blogosphere. The app most widely used for searching (Blog = 30/52; 57.69%) was Google, that for reproduction (Blog = 7/52; 13.46%) was YouTube, and that for production and editing (Blog = 29/52; 55.77%) was PowerPoint. Production and editing were also accomplished using the tablet’s default apps, such as the camera and video and image editing tools. The blogs cited other multimedia apps for production and editing, such as Vivavídeo, Wevídeo or Videoshow; the blogs also reveal a pattern of innovative activities emerging in classes in the natural sciences.

The study also found a typology of narration-communication activities developed for use on the tablet that only appeared in the blogs (Blog = 14/52; 26.92%). These activities were mainly activated by interactive apps (Blog = 14/52; 26.92%) such as eTwinning—Twinspace and Padlet; other apps were used for reproduction (Blog = 9/52; 17.31%), such as Qr Droid Private (to read and reproduce QR codes). One particular case, using eTwinning—Twinspace, showed how a community of school centers across Europe had communicated and developed an educational project with other schools in Spain. The Padlet app enabled the children to share multimedia resources and information from a range of sources online.

The pattern of activities associated with audiovisual/multimedia creation was mainly evident in natural sciences classes (GD = 55/120, 45.83%; Blog = 26/52, 50%). This praxis involved the recording/editing of videos and/or the presentation of class content. The children usually used the tablet’s default apps (camera, video and image editor) that allowed them to perform production and editing tasks (GD = 51/120, 45.50%; Blog = 24/52, 46.15%); content reproduction (GD = 28/120, 23.33%; Blog = 11/52, 21.15%) was accomplished using PowerPoint, for example. The blogs also discussed other apps, such as Prezi and Canva, which enabled the children to create presentations and post them online. In one blog, the writer reports that the children created audiovisual/multimedia content for a project on volcanoes using a range of apps, which included Phoster (poster production), Flowboard (creating presentations with added multimedia effects), Keynote (designing presentations), Skitch (snapping and editing screen grabs) and Book Creator (producing an ibook in ePub format).

The natural sciences are special in that it generated a typology of practices around graphic composition or drawing (GD = 30/120, 25%; Blog = 15/52, 28.85%). The children created work involving icons for different thematic areas of the subject. In some cases, the tablet’s default editing tools were used for such activities (Android or iOS), or the Fresh Paint app (GD = 30/120; 25%); the blogs also mentioned the use of image and video editing tools (Blog = 15/52; 28.85%) such as PicCollage and StoryBoard.

The more traditional typology of activities supported by the tablet was also evident at most of the school centers analyzed in this study, whereas those described in teachers’ blogs were more innovative. This latter is particularly encouraging for children’s educational processes, as it boosts their meaningful learning. The more traditional practices for the tablet included the identification and relation of academic concepts and the reconstruction of mutilated texts. These activities are common both in the natural sciences (GD = 75/120, 62.50%; Blog = 19/52, 36.54%) and in mathematics (GD = 45/120, 37.50%; Blog = 11/52, 21.15%) praxis, though less so in mathematics. For example, the children carried out exercises that required them to recognize operations that expressed correct or incorrect calculations, or related various units of measurement, etc.

In the classrooms analyzed in this study, it was found that students in mathematics used apps developed by the publishers of the most widely used mathematics course books in Spain (Anaya, Edelvives and Salvia) (GD = 26/120; 21.67%). These apps provided didactic activities for all levels of primary education; the following extracts are an example of the use of these apps on the tablet in activities employing a more traditional methodology (School Center nº 1, GD nº 5, students in 5th grade, aged 10–11, lines 24–33):


*Participant 5: In Maths, we do activities to solve mathematical problems.*



*Participant 2: For example, we used the Edelvives app. It gives us a problem and we have to solve it. And to do that, we have to indicate which option is correct from the alternatives it gives us (a, b, c and d).*



*Participant 3: In other cases, there is a space where we put the answer but they don’t give us any options.*



*Participant 4: And other examples, where we have to connect measures, for example units, tens, hundreds.*



*Participant 1: We also do this in exercises to link geometric figures to their definition.*


In mathematics, the main practices developed for students related to numerical operations, which was evident in the praxis followed in the classrooms analyzed (GD = 50/120; 41.67%) and in the deliberations on the blogs (Blog = 21/52; 40.38%). Problem solving and arithmetical operations were the most common activities for students in the school centers observed, with the apps most commonly used on the tablet being those provided by Edelvives, Salvia and Anaya (for the reproduction, production and editing, systematization and synthesis of course content). Blogs that presented mathematics activities for use in the classroom described tasks such as selecting geometrical figures, number riddles, etc., across the range of primary education levels.

### 3.2. Curricular Content Promoted by Tablet Use (RQ3)

This section deals with the activities undertaken by the primary education students in relation to the curricular content they assimilated. The work done by children in the natural sciences and mathematics across all levels of primary education emphasizes the development of a range of curricular content, which was evident in the seven patterns of practices detected in this study (Table 3).

The analysis showed that the typology of practices associated with search/gathering information in the natural sciences enabled the children to acquire content relating to living beings (GD = 58/120, 48.33%; Blog = 19/52, 36.54%), to the environment (GD = 16/120, 13.33%; Blog = 19/52, 36.54%) and to states of matter (GD = 31/120, 25.83%; Blog = 3/52, 5.77%). The tablet facilitated the children’s understand of the topics through multimedia that graphically illustrated the properties and states of matter, as highlighted by the following extracts (School Center nº 3, GD nº 3, children in the third grade of primary education, aged 8–9, lines 8–14):


*Participant 5: We do a lot of activities in which we search for information.*



*Participant 3: Especially in Natural Sciences.*



*Participant 4: At the moment, we are working on states of matter, and so we search for information on them.*



*Participant 1: We do this on Google on the tablet. We look for information and for images, too, on Google.*



*Participant 2: We search for images and examples that can help us to see the states of matter.*


In other cases, the school activities described in the blogs refer to the development of content on the ecological garden—for example, using the tablet to gather data on constructing a vegetable patch—and on types of animals (vertebrates and invertebrates) and food (using icons to design a food pyramid).

The children studied content on living beings (GD = 52/120, 43.33%; Blog = 11/52, 21.15%) and states of matter (GD = 23/120, 19.17%; Blog = 8/52, 15.38%) through practices focused on the identification and relation of concepts and mutilated text completion. These practices broadened their learning on plant composition and typology and on the characteristics of animals; the use of graphic activities on the tablet helped to reinforce this knowledge acquisition. Other activities performed by the children related to environmental sustainability and the ecosystem. However, the pattern of activities that emerges from the implementation has little to do with narration and communication; this aspect only appears briefly in the blogs, in content on the environment and living beings (Blog = 8/52, 15.38%; 6/52, 11.54%, respectively).

Other typologies of activities supported by the tablet in the natural sciences and, to a less extent, in mathematics, promoted learning through discovery, exploration and enquiry as part of the development of class content. This enabled students to deepen their understanding of topics relating to living beings (GD = 27/120, 22.50%; Blog = 13/52, 25%). In the classroom cases analyzed, the praxis applied to tablet use enabled us to identify the implementation of content associated with states of matter (GD = 13/120; 10.83%), while the study of the blogs showed the development of activities on the environment, for example, in sustainability and the ecosystem (Blog = 16/52; 30.77%).

In mathematics, the children worked on practices that helped them to acquire content about measurements of capacity and volume and equivalences between bank notes and coins. The singularity of mathematics meant that the content was developed mainly around numbers (GD = 50/120, 41.67%; Blog = 17/52, 32.69%), including mathematical operations with natural numbers, fractions, etc., and terms related to measurements (Blog = 4/52; 7.69%), such as units of length, weight, etc. There was also some content that had been developed for geometry (GD = 5/120; 4.17%) based on graphics of geometrical figures.

## 4. Discussion

The activities developed for tablet use for primary education students in the natural sciences and mathematics are not very innovative or ludic. The pattern of practices most commonly developed in the natural sciences is the search for and gathering of information, which clearly demonstrates the potential of the Information Society to apply technologies to access, use and share content in multiple areas [41,42,43]; however, research has shown that the teaching-learning process in the Sciences continues to adopt a traditionalist approach that limits students’ opportunities for meaningful learning [44].

The analysis of the educational blogs revealed a pattern of activities developed for children that were overwhelmingly based on searching for information, with a smaller set of activities dedicated to more ludic tasks, such as drawing, photography and making short videos. The most commonly used app on the tablet was the Google search engine, which accounts for more than 65% of all Internet searches worldwide, way ahead of Bing (13%), Baidu and Yahoo (9% respectively) [45]. Tablet-based tasks can boost students’ globalized learning, in line with Constructivist theory [46], and favor the use of gamified strategies in education.

The class content on living beings and states of matter in the natural sciences was mainly developed by the children following patterns of practice that related to the search for, and gathering of, information. Other practices promoted student learning through discovery, exploration and enquiry in this subject. The app most commonly used on the tablet in this context was YouTube, which added a ludic aspect to learning about the topics of living beings, the environment and states of matter. Enquiry as a teaching-learning strategy in the Sciences is opening up innovative and alternative forms of study, and helping to achieve better learning outcomes; teaching methodology should take inspiration from the steps in the scientific process [47].

A more innovative and ludic profile was identified in practices described in the blogosphere than in the case studies analyzed. Blogs being a digital, mass-diffusion medium means the emphasis is on the maximum visualization of activities that are interesting, impactful or more innovative. Blogs have proliferated in recent years on social media, and research into their content has increased [48,49]. Blogs have become one of the most common data platforms [50] and are considered an online platform [51]. Blogs are also excellent settings for the exchange, diffusion and transfer of all types of experiences.

The analysis of praxis in this study yielded a pattern of practices for audiovisual and multimedia creation. The content that students worked on in the activities using apps, for the production and editing of content, was mainly to do with living beings and the environment. The use by students of the tablet’s default camera and editor, with apps such as Vivavídeo and Webvídeo, was common. This type of app is interesting for the ludic input of the multimedia format, as it stimulates the children’s imagination and stretches their thinking capacity, echoing the effect of gamification in the classroom [52,53]. This emphasizes the motivational potential of videogames by transferring game design elements to settings far removed from gaming [54].

In mathematics, the tablet is lamentably underused in support of the learning processes of children in primary education. The pattern of practices consisted of performing numerical operations and identifying and relating concepts based on a typically traditional methodology. The few digital apps available are those provided by course book publishers in which the children solve basic mathematical problems (adding, subtracting, multiplying and dividing). The content the children work on is mainly numerical, and yet use of the tablet can succeed in motivating children in mathematics [55,56]. For this to happen more consistently, methodological strategies need to be developed that arouse children’s interest in order to generate a positive impact on their learning [57]. Such methodologies need to be active or gamified to motivate children, reinforce their learning and improve their academic performance.

The content that children work on in the natural sciences and mathematics supported by the tablet, taking into account Spanish education legislation [4], is still a small part of the curriculum. In the natural sciences, the content developed for the tablet only relates to living beings, states of matter and energy, while, in mathematics, children work on content relating to numbers, units of measurement, geometry and statistics, but do not use the tablet to work on aspects of the course content such as processes, methods and attitudes in mathematics, or probability.

Future teachers must know how to use ICT to innovate in schools. Unfortunately, sometimes curricula are not up to date, and it is necessary to renew their content, focusing on methodologies that achieve meaningful learning from the school curriculum. This problem is present in several countries, including some of the most developed, so it is interesting to think about how to solve the deficiencies. In this sense, it is essential that the curriculum for teacher training in the subjects of natural sciences and mathematics teaches and promotes the development of innovative practices, in which students are actively involved [58,59].

These future teachers must learn to develop playful practices with children using digital devices such as tablets to promote experiential and meaningful learning. To this end, it would be interesting for the education students themselves, in their teacher training, to use different innovative methodological strategies that include the tablet for the development of new educational initiatives (flipped classroom, gamification, project-based learning...). In addition, continuous training is needed for in-service teachers to help them improve their use of active methodologies with technological resources (such as the tablet and the use of digital educational applications) to address the specific contents of the school curriculum, particularly in the subjects of natural sciences and mathematics [60].

On the positive side, it should be noted that there has been a rapid increase in the number of apps made for use with tablets since they were first introduced into the classroom [61]. App development is undergoing an unprecedented evolution, in particular in regard to social and educational apps, and their usage is making knowledge more widely available than ever, offering countless opportunities to generate and share it [62]. In mathematics, the development of apps such as GeoGebra [63] for the tablet enables children to learn about abstract concepts of geometry, algebra and calculus more easily and in a more motivational way by fomenting greater interaction with and combination of elements. The tablet can store a wide variety of ludic apps that can be used to implement a wide range of activities for children to use both in the classroom and at home.

## 5. Conclusions

The work done by children using the tablet is characterized by a variety of patterns of activities. In this study, the activities that primary education students carried out on the tablet in natural sciences and mathematics did not greatly boost innovation in education, yet a trend towards innovation and ludic practices in tablet-based activities in natural sciences was observed. The activities developed for the tablet in natural sciences far outnumbered those for mathematics. The typology of praxis most commonly seen was the search for and gathering of information, with the Google app, for the study of living beings. Many activities were dedicated to developing content on the environment that stimulates children’s learning through discovery, exploration and enquiry, mainly using YouTube.

In mathematics, the praxis was predominantly traditional, based on course book exercises that work on content to practice numerical operations. However, despite the rare usage of the tablet for mathematics in primary education, some apps show sufficient potential to motivate children in the learning of mathematics.

## 6. Findings and Study Limitations

This study broadens the knowledge and understanding of the didactic potential of tablet use by children in primary education connected with the subjects of the natural sciences and mathematics. Its main contribution is the identification of key patterns in the types of activities developed for tablet use, and of the potential uses for certain apps created for tablets in the development course content among young children.

The main limitation of this work could be the small number of classes and participants analyzed in this study, and the typology of the activities supported by the tablet in the school centers. However, the study includes a set of participants who are largely heterogeneous (representing four of the six stages of primary education). Future research could extend the study of blogs undertaken in this work, as a source of activities supported by the tablet for use in the natural sciences and mathematics. It would also be interesting to extend the analysis to other school subjects in the primary education curriculum where the tablet is, or could be, used.

## Figures and Tables

**Figure 1 children-10-00250-f001:**
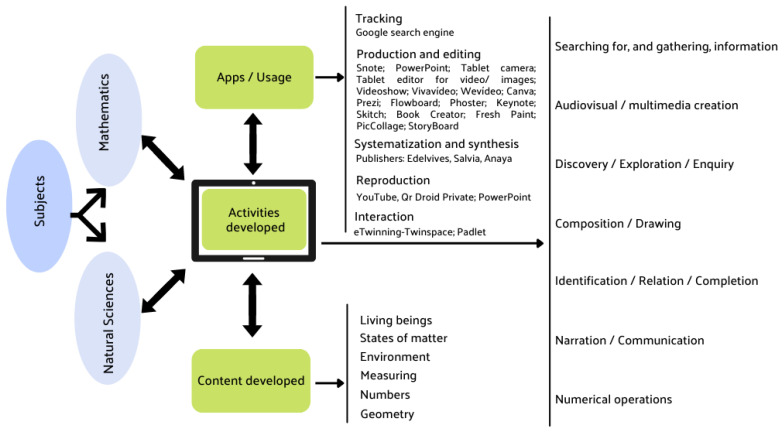
A general view of the categorization obtained.

**Table 1 children-10-00250-t001:** Categories established for content analysis.

Categories (1st Level and Definition)	Type of Activities Done with Tablet (the Various Educational Practices Implemented Using the Tablet)							Apps and Usage on the Tablet (the Apps Identified, and How They Are Used, on the Tablet)					Typology of Content with the Tablet (Content Created by the Children Based on Practices Developed on the Tablet)					
**Codes** (2nd level)	Searching for, and gathering, information	Audiovisual/Multimedia creation	Discovery/Exploration/Enquiry	Composition/Drawing	Identification/Relation/Completion	Narration/Communication	Numerical operations	Tracking (Google search engine)	Production and editing (Snote; PowerPoint; Tablet camera; Tablet editor for video/images; Videoshow; Vivavídeo; Wevídeo; Canva; Prezi; Flowboard; Phoster; Keynote; Skitch; Book Creator; Fresh Paint; PicCollage; StoryBoard)	Systematization and synthesis (Publishers: Edelvives, Salvia, Anaya)	Reproduction (YouTube; QR Droid Private; PowerPoint)	Interaction (eTwinning-Twinspace; Padlet)	Living beings	States of matter	Environment	Measuring	Numbers	Geometry
**Definition** (2nd level)	Activities based on searching for, and compiling, information.	Practices centered on audiovisual and multimedia production.	Actions that generate learning through discovery, exploration and enquiry.	Practices associated with composition and/or drawing for different study topics.	Activities centered on identifying, relating and completing ideas/concepts on topics in the curriculum.	Practices of narration, composition and communication in relation to aspects of the topics in the curriculum.	Activities involving numerical operations: arithmetic, measurement, etc.	Use of apps for exploring information.	Use of the app for producing and editing text or multimedia content.	Use of the publishers’ apps for storing, organizing, grouping and specifying learning content.	Use of the app for reproducing text or multimedia content.	Use of the app for developing communicative interaction.	Content created by practices developed with the tablet on animals, plants, food, the muscles, photosynthesis, etc.	Content created by practices on the tablet on raw materials, their properties, states, etc.	Content created from practices on the tablet on the environment: its protection, ecosystems, ecological garden, sustainability, etc.	Content created from practices on the tablet on measuring, size, weight, capacity, units of time and measurement.	Content created from practices with the tablet involving operations with natural numbers and fractions.	Content generated from practices with the tablet on figures involving plane and space.

**Table 2 children-10-00250-t002:** Typology of activities with the tablet in the natural sciences and mathematics.

	Categorization	Activity with the Tablet (1st Level)			Apps/Usage (1st Level)		
		Code (2nd Level)	GD f (%)	Blog f (%)	Code (2nd Level)	GD f (%)	Blog (f (%)
**Subject in the school curriculum**	Natural Sciences	Search/ Gathering Information	105 (87.5)	41 (78.85)	Search (Google) Production/editing (Snote) Reproduction (YouTube)	105 (87.5) 15 (12.50) 31 (25.83)	41 (78.85) 7 (13.46) Ø
	Mathematics		Ø	6 (11.54)	Search (Google) Production/editing (PowerPoint)	Ø	6 (11.54%) 2 (3.85%)
	Natural Sciences	Audiovisual/Multimedia Creation	55 (45.83)	26 (50)	Search (Google) Production/editing (Tablet camera; Tablet video/image editor; Prezi; Canva; Flowboard, Phoster; Keynote; Skitch; Book Creator) Reproduction (PowerPoint)	12 (10) 51 (45.50) 28 (23.33)	2 (3.85) 24 (46.15) 11 (21.15)
	Mathematics		Ø	2 (3.85)	Search (Google) Reproduction (YouTube)	Ø	2 (3.85) 2 (3.85)
	Natural Sciences	Discovery/Exploration/Enquiry	40 (33.33)	30 (57.69)	Search (Google) Production/editing (PowerPoint); Tablet camera; Tablet video/image editor; Vivavídeo; Wevídeo/Videoshow Reproduction (YouTube)	40 (33.33) 36 (30) 16 (13.33)	30 (57.69) 29 (55.77) 7 (13.46)
	Mathematics		Ø	4 (7.69)	Search (Google) Production/editing (Canva) Reproduction (YouTube)	Ø	4 7.69) 4 (7.69) 3 (5.77)
	Natural Sciences	Composition/Drawing	30 (25)	15 (28.85)	Search (Google) Production/editing (Snote; Tablet video/image editor; Fresh Paint; PicCollage; StoryBoard. Reproduction (YouTube)	5 (4.17) 30 (25) 18 (15)	3 (5.77) 15 (28.85) Ø
	Mathematics		Ø	Ø	Ø	Ø	Ø
	Natural Sciences	Identification/Relation/Completion	75 (62.50)	19 (36.54)	Production/edition (PowerPoint) Reproduction (YouTube) Interaction (Padlet)	35 (29.17) 40 (33.33) Ø	10 (19.23) Ø9 (17.31)
	Mathematics		45 (37.50)	11 (21.15)	Production/editing (PowerPoint) Reproduction (YouTube) Systematization and synthesis (publishers: Edelvives; Salvia; Anaya) Interaction (Padlet)	19 (15.83) 19 (15.83) 26 (21.67) Ø	5 (9.62) Ø Ø 6 (11.54)
	Natural Sciences	Narration/Communication	Ø	14 (26.92)	Production/editing (VivaVídeo) Reproduction (Qr Droid Private) Systematization and synthesis (publishers: Edelvives; Salvia; Anaya) Interaction (eTwinning-Twinspace; Padlet)	Ø	7 (13.46) 9 (17.31) 2 (3.85) 14 (26.92)
	Mathematics		Ø	Ø	Ø	Ø	Ø
	Natural Sciences	Numerical Operations	Ø	1 (1.92)	Reproduction (YouTube)	Ø	1 (1.92)
	Mathematics		50 (41.67)	21 (40.38)	Production/editing (Tablet video/image editor; Canva) Reproduction (YouTube) Systematization and synthesis (publishers: Edelvives; Salvia; Anaya)	12 (10) 28 (23.33) 10 (8.33)	11 (21.15) 10 (19.23) Ø

Legend: GD = Group discussion. f (%) = Absolute frequency (percentage).

**Table 3 children-10-00250-t003:** The content studied based on practices developed with the tablet.

	Categorization	Activity with the Tablet (1st Level)	Type of Content Developed with the Tablet (1st Level)		
		Code (2nd Level)	Code (2nd Level)	GD f (%)	Blog f (%)
**Subject**	Natural Sciences	Search/Gathering information	Living beings Environment States of matter	58 (48.33) 16 (13.33) 31 (25.83)	19 (36.54) 19 (36.54) 3 (5.77)
	Mathematics		Measurement	Ø	6 (11.54)
	Natural Sciences	Audiovisual/Multimedia creation	Living beings Environment States of matter	32 (26.67) 9 (7.50) 14 (11.67)	5 (9.62) 19 (36.54) 2 (3.85)
	Mathematics		Measurement	Ø	2 (3.85)
	Natural Sciences	Discovery/Exploration/Enquiry	Living beings Environment States of matter	27 (22.50) Ø 13 (10.83)	13 (25) 16 (30.77) 2 (3.85)
	Mathematics		Measurement	Ø	4 (7.69)
	Natural Sciences	Composition/Drawing	Living beings Environment Material	19 (15.83) 11 (9.17) Ø	19 (36.54) 13 (25) 2 (3.85)
	Mathematics		Ø	Ø	Ø
	Natural Sciences	Identification/Relation/Completion	Living beings States of matter	52 (43.33) 23 (19.17)	11 (21.15) 8 (15.38)
	Mathematics		Numbers Measurement Geometry	26 (21.67) 13 (10.83) 5 (4.17)	9 (17.31) 2 (3.85) Ø
	Natural Sciences	Narration/Communication	Environment Living beings	Ø Ø	8 (15.38) 6 (11.54)
	Mathematics		Ø	Ø	Ø
	Natural Sciences	Numerical operations	Measurement	Ø	1 (1.92)
	Mathematics		Numbers Measurement	50 (41.67) Ø	17 (32.69) 4 (7.69)

Legend: GD = Group Discussion. f (%) = Absolute frequency (percentage).

## Data Availability

The data presented in this study are available from the corresponding author on reasonable request.

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
