# Peer review of "Tablet-Based Praxis Developed for Children in Primary Education Studying Natural Sciences and Mathematics"

_children, 2023, doi:10.3390/children10020250_

Round 1

Reviewer 1 Report

The Authors presented a very interesting results of research on the use of tablets in teaching natural sciences and mathematics in primary education in Spain.

The literature analysis is correct. In terms of methodology, the research was properly planned and conducted.

The description of the research procedure used is very good. The Authors used qualitative research, a narrative-ethnographic approach.

It should be noted that the analysis of the research results is very detailed.

The discussion was well crafted.

The only things I would like to ask the Authors are: what are their suggestions regarding the introduction of exercises using various applications on a tablet to the curriculum for teachers of natural sciences and mathematics?

Do the Authors consider it necessary to teach future teachers how they should use tablets in primary school? Could these changes also be introduced in the curricula of future teachers of natural sciences and mathematics in other countries?

My questions may be answered in the conclusions.

The text is very interesting, it does not require major editorial corrections.

I recommend the text for publication.

Reviewer 2 Report

Dear authors

The presented topic has a great potential in educational research field. Results also contribute to educational quality improvement to the extent that they reveals actual educational practices and promote a diagnosis of the educational reality revelling weaknesses that can be solved but also some good practices. They arise from a dense scientific investigation, well implemented, witch phases are well described (theoretical basis, problems, objectives, methodology, data collection instruments and participants).
I highlight the data analysis process, whose evidence demonstrates the rigor with which it was conducted. There is a vast evidence of data analysis. You concluded a content analysis process that results in a categories and subcategories scheme that makes it possible to understand how collected data respond to the objectives of the investigation.

Aspects to improve.

On line 28 I suggest "at home" instead of "in the home".

In line 56 and 57 I don't understand what is meant by "Education in all things scientific in children". Improve it please.

Conclusion

ORIGINALITY. The article is innovative enough to justify its publication. It addresses relevant educational scenarios, based on a dense conceptual and theoretical approach.

CONTRIBUTION TO SCIENTIFIC KNOWLEDGE. TEXT ORGANISATION. The article contributes to scientific knowledge, as it presents a clear and logical organization and important results. Additionally, conclusions are strictly drawn from concrete and verifiable evidence. It is a valuable example in terms of research methodology implementation.

WRITING AND LANGUAGE. The ideas are presented in a sequential and linked way. It follows a logical scientific article structure according with research questions.

ARGUMENTS. There aren’t failures of coherence and internal consistency that compromise the relevance of its content and the validity of the presented results. Theoretical basis is dense (recent multidisciplinary publications with scientific relevance).  

TITLE. The title matches the content.  

METHODOLOGY. The research methodology is clearly described. There weren’t detected gaps that compromise methodological rigor, consequently the relevance and validity of the results. The description of the research context, the associated problem and the research objectives, are clearly enough. Qualitative data is methodically analysed.

RESULTS. The validity of the results are supported by the of theoretical foundation and by a clear and explicit methodological design.  Research findings are clearly explained.
